# Real-Time MR-Guided Lumbosacral Periradicular Injection Therapy Using a 0.55 T MRI System: A Phantom Study

**DOI:** 10.3390/diagnostics15111413

**Published:** 2025-06-02

**Authors:** Saher Saeed, Jan Boriesosdick, Arwed Michael, Nina Pauline Haag, Julian Schreck, Denise Schoenbeck, Matthias Michael Woeltjen, Julius Henning Niehoff, Christoph Moenninghoff, Jan Borggrefe, Jan Robert Kroeger

**Affiliations:** Department of Radiology, Neuroradiology and Nuclear Medicine, Johannes Wesling University Hospital, Ruhr University Bochum, 44801 Bochum, Germany; jan.boriesosdick@muehlenkreiskliniken.de (J.B.); arwed.michael@muehlenkreiskliniken.de (A.M.); ninapauline.haag@muehlenkreiskliniken.de (N.P.H.); julian.schreck@muehlenkreiskliniken.de (J.S.); matthiasmichael.woeltjen@muehlenkreiskliniken.de (M.M.W.); julius.niehoff@muehlenkreiskliniken.de (J.H.N.); christoph.moenninghoff@muehlenkreiskliniken.de (C.M.); jan.borggrefe@muehlenkreiskliniken.de (J.B.); janrobert.kroeger@muehlenkreiskliniken.de (J.R.K.)

**Keywords:** MR-guided intervention, periradicular therapy, low-field MRI, spinal injections, phantom study

## Abstract

**Objective:** The purpose of this study was to evaluate the accuracy and feasibility of magnetic resonance (MR)-guided periradicular nerve root injection therapy (PRT) using a 0.55 T magnetic resonance imaging (MRI) system with fast dynamic imaging in a phantom. **Methods:** Five radiologists with varying levels of experience in PRT performed nine randomly assigned PRT procedures: three under MR guidance, three under CT guidance using a fully integrated laser navigation system, and three under conventional CT guidance, all on a specialized phantom of the lumbar spine. The PRTs were assessed by two experienced neuroradiologists with expertise in interventions, using a scale of 1–5, as follows: 5 = excellent to very good, 4 = good, 3 = satisfactory 2 = bad, 1 = very bad. The puncture time and total intervention time were noted. **Results:** All procedures were technically successful. The subjective evaluation of the PRTs showed similar results with a median of 5 for all three guidance systems. Additionally, there was no significant difference with respect to pure puncture time (the period after needle path determination) among all PRTs (Mean ± SD): MR-guided 178 ± 117 s, CT-guided with laser system 186 ± 73 s, and the conventional CT-guided 218 ± 91 s (*p* = 0.482). However, the total procedure time including planning images was significantly higher for MR-guided PRT (700 ± 182 s) compared to CT guidance with laser system (366 ± 85 s) and conventional CT guidance (358 ± 150 s; *p* = 0.012). **Conclusions:** Real-time MRI-guided lumbosacral periradicular injection therapy utilizing a 0.55 T MRI system is feasible with similar puncture times to CT guidance but consumes more intervention time due to the duration of planning sequences. Limitation: The study utilized a stationary phantom made of homogeneous material, which provides an incomplete representation of real tissue properties and motion complexity applied to human beings.

## 1. Introduction

Chronic low back pain is frequently diagnosed in developed countries. The socioeconomic burden of this condition is substantial, with an estimated global prevalence of 9%, which also makes it one of the primary causes of disability [1,2].

Most patients suffer from acute or chronic disk herniation, degenerative spinal disease, or postoperative epidural fibrosis. Conservative treatment options, including oral pain relievers and physical therapy, can relieve pain in most patients. However, conservative treatment fails in a relevant share of patients [3,4]. The selective injection of corticosteroids and anesthetics near the nerve roots has proven to be a useful tool in the diagnosis and treatment of radicular pain in patients’ refractory to conservative treatment options. [5,6,7].

Up to the present, fluoroscopy and computed tomography (CT) continue to be the primary modalities employed for guiding percutaneous spinal injection procedures. Fluoroscopy provides real-time guidance and exposes patients to lower levels of radiation compared to CT scans. However, its accuracy is somewhat limited in contrast to CT imaging. CT scans offer superior spatial resolution and clearer differentiation between bone and soft tissue, thereby ensuring more precise injections.

Because of concerns regarding ionizing radiation, a step-by-step approach is usually employed with CT guidance rather than using CT-fluoroscopy. However, even when contrast material is used to confirm needle placement, the visualization of soft tissue remains suboptimal [8,9,10].

The development of MR-compatible therapy needles now allows these interventions to be performed under MR imaging guidance. MR-guided interventions can be performed on most clinical MRI whole-body scanners, whereby a wide bore opening or even an open configuration is advantageous. The diagnostic benefits of interventional magnetic resonance imaging (MRI) over CT and fluoroscopy include high soft tissue contrast, unrestricted multiplanar imaging capabilities, and the absence of ionizing radiation.

The purpose of this study was to evaluate the accuracy and feasibility of magnetic resonance (MR)-guided periradicular nerve root injection therapy (PRT) using a 0.55 T magnetic resonance imaging (MRI) system with fast dynamic imaging in a phantom.

## 2. Methods

### 2.1. Study Design

The current study is a prospective comparative analysis utilizing a phantom model compatible with both CT and MRI imaging of the lumbosacral region. It aims to assess operators’ performance across various experience levels in percutaneous periradicular injection therapy procedures (PRTs). Specifically, the study compares PRTs guided by MRI (MR group) against those guided by a complete laser system with CT (CT-L group) and the standard CT method (CT).

In this study, five radiologists with varying levels of expertise in the field participated. Two radiologists had extensive experience, each having conducted more than 100 PRT procedures. Additionally, two radiologists had conducted more than 50 PRTs, signifying a moderate level of experience. Finally, one radiologist had performed more than 25 PRTs, indicating a lower level of experience. All of the previous procedures had been performed using CT guidance, and none of the radiologists had previous experience with MR guidance.

These radiologists collectively performed a total of nine PRT procedures, which were methodically distributed amongst three categories. Each category was characterized by a different guidance system, including magnetic resonance (MR) guidance, computed tomography (CT) guidance supported by an integrated laser navigation system, and conventional CT guidance.

The PRT targets for each procedure were randomly assigned. The procedures were meticulously performed on a specialized phantom model, which served as a surrogate to replicate the complexities encountered in clinical practice.

As this was a feasibility study with a predefined number of procedures and operators, no formal power analysis was conducted. The study was not designed to detect statistically significant differences with high power but rather to provide an initial comparative evaluation of guidance modalities in a controlled phantom setting. Future studies with larger sample sizes and formal sample size estimation will be needed to confirm these findings and explore subtle performance differences.

### 2.2. Phantom of Lumbosacral Spine: Lumbar Training Phantom, CIRS Model 034 by Sun Nuclear

CIRS Model 034 Lumbar Training Phantom (Sun Nuclear Corporation, Melbourne, FL, USAis a specialized tool designed for practicing spinal procedures under fluoroscopic guidance. It mimics lifelike anatomy to enhance hand–eye coordination and proficiency in various spinal procedures like sacroiliac joint injections, epidurals, diskography, nerve blocks, and facet blocks. Key features include realistic needle resistance, a self-sealing puncture membrane for durability, comprehensive anatomical representation, including key ligaments and vertebral levels, compatibility with multiple imaging modalities, and the ability to train for six different spinal procedures.

### 2.3. MRI Needle Guidance

The phantom was placed in the supine position in a 0.55 Tesla whole-body MRI system (Magnetom Free.Max, Siemens, Erlangen, Germany), which provides an 80 cm scanner opening beneficial for MR-guided needle interventions. (Figure 1)

After generating a localizer, a standard T2-weighted turbo spin echo (TSE) axial sequence was acquired for access planning. The main scan parameters of this sequence were as follows: TE, 106 ms; TR, 2600 ms; flip angle, 160°; bandwidth, 100 Hz/pixel; number of averages, 3; slice thickness, 4 mm, sagittal localizer confirmed the correct level to be infiltrated. BEAT IRT TRUFI was used to determine the exact skin entry point using the finger-pointing technique, and the needle guidance had the following parameters: TE, 2.67 ms; TR, 749.1 ms; flip angle, 90°; bandwidth, 668 Hz/pixel; number of averages, 1; slice thickness, 8 mm; acquisition plane, axial and sagittal.

An MRI-compatible 20-G needle, (ITP) Innovative Tomography Products GmbH, Bochum, Germany, KIM-22/10 Length: 100 mm Diameter: 22 G (0.70 mm) Standard Tip, was inserted from a dorsolateral angle. The BEAT IRT TRUFI sequence was used to monitor the needle path from the skin entry to the target anatomy in real time. The operator had access to real-time imaging using a monitor in the scanner room. (Figure 2)

The same Standard T2-weighted turbo spin echo (TSE) axial sequences used for planning was immediately performed to confirm the position of the needle tip.

### 2.4. 3D Laser MDCT Guidance System

The 3D laser guidance system (myNeedle Companion and myNeedle Laser, Siemens Healthineers, Erlangen, Germany) consists of four specialized laser modules positioned at the 9, 11, 1, and 3 o’clock locations around the scanner gantry (Somatom X.ceed, Siemens Healthineers, Germany; see Figure 3). The system integrates with a proprietary planning software (myNeedle Guide, Siemens Healthineers, Germany), which enables the simultaneous configuration of multiple needle trajectories, whether in-plane or with complex angulation. Once a trajectory is defined, the system automatically adjusts the patient table to an optimal position for visual access.

The laser system then projects the selected needle pathway directly onto the intervention field. A central laser crosshair identifies the designated entry point, while the crossing beams of two fan-shaped lasers represent the needle insertion angle. The operator advances the needle along the projected trajectory, as depicted in Figure 4 and Figure 5.

### 2.5. Conventional Needle Guidance

Following the acquisition of the CT volume dataset of the phantom, the optimal needle paths were planned. The insertion site was determined by measuring the lateral offset from the scanner’s midline and marking the location on the phantom using alignment lasers and a standard measuring tape. The needle was then gradually inserted into the phantom, with its trajectory adjusted based on serial CT scans to ensure accurate targeting. Corrections were made iteratively until the desired endpoint was reached. Both the conventionally guided PRTs and those guided by the 3D laser system were conducted using the same multidetector CT unit (Somatom X.ceed, Siemens Healthineers, Germany), utilizing identical imaging protocols. Spiral CT was performed with a slice thickness of 3.0 mm, a tube voltage of 100 kV, and an effective tube current of 241 mAs. For sequential scanning, parameters included a 0.6 mm slice thickness, 130 kV tube voltage, and 128 mAs effective current (Figure 6).

Total procedure times were analyzed, and puncture times were measured from the time the needle path was determined until the user finished placing the needle.

The assessment of PRT processes was conducted by two skilled neuroradiologists with extensive expertise in interventional radiology, each with over 20 years of experience in the field. They used a scale that ranged from 1 to 5, with each number representing a particular level of performance, as follows: A score of 5 was given to procedures that displayed an excellent level of quality, indicating outcomes that were excellent or very good. Procedures meeting a high-quality standard rated “good” at 4. Procedures that met the necessary expected standard satisfactorily received a rating of 3, indicating “satisfactory” results. Procedures falling below the required standard were given a rating of 2, denoting “bad” outcomes. Procedures critically evaluated as “very bad” received the lowest rating of 1.

## 3. Statistic

Values are given as median (interquartile range) or mean ± standard deviation. Significance of differences between groups was tested using the Kruskal–Wallis test by rank for ordinal variables (subjective rating) and by one-way ANOVA for continuous variables (puncture time and distance from needle tip to the nerve root). A pair-wise post hoc testing with Bonferroni-corrected *p*-values was performed. *p*-values of <0.05 were regarded as statistically significant.

## 4. Results

### 4.1. Technical Success

All procedures were successfully completed, demonstrating the overall dependability of the three guidance systems.

### 4.2. Subjective Evaluation

The subjective evaluation of the quality of needle positioning showed good results for all three guidance systems (MR guided: 5 (0), CT-guided with laser system: 5 (0) and conventional CT-guided: 5 (1)). However, upon statistical analysis, significant differences between the three groups were identified (*p* = 0.031), and post hoc testing revealed a significant difference between MR guidance and conventional CT guidance (*p* = 0.031) and higher mean score for MR guidance (4.9 ± 0.3 vs. 4.5 ± 0.7). The score distribution is shown in Figure 7.

### 4.3. Distance from Needle Tip to the Nerve Root

The measured distance from the needle tip to the nerve root was not significantly different between groups (MR guidance: 1.8 ± 1.4, CT guidance with laser system: 2.8 ± 2.6, conventional CT guidance: 3.1 ± 3.6, *p* = 0.402).

### 4.4. Puncture Time

No significant difference was found for the puncture time between groups (MR guidance: 178 ± 117 s, CT guidance with laser system: 186 ± 73 s, conventional CT guidance: 218 ± 91 s (*p* = 0.482). These results are visualized in Figure 8.

### 4.5. Total Procedure Time

The total procedure time, including planning images, exhibited variations across the three guidance systems, as follows: MR-guided: 700 ± 182 s, CT-guided with laser system: 366 ± 85 s, and conventional CT-guided: 358 ± 150 s (*p* = 0.012).

A significant difference was observed in the total procedure time among the three guidance systems, with MR-guided PRTs taking significantly longer than the other two systems (Figure 9).

## 5. Discussion

In our comprehensive analysis of a series of fifteen MRI-guided PRT procedures in a phantom, we achieved a consistently high level of technical procedural success in all cases. Additionally, our results show no statistically significant difference in the subjective evaluation of the quality of needle positioning between MRI-guided PRTs and those performed under CT guidance or in combination with CT and laser system, indicating that each method facilitated accurate needle placement with proficiency.

Numerous studies have shown that periradicular nerve root injection therapy, guided by imaging techniques, is a safe and effective treatment for patients who find conventional treatments unsatisfactory and for whom traditional surgery is not recommended [9,10,11,12,13].

X-ray fluoroscopy and computed tomography are frequently used to perform percutaneous interventions in pain therapy. The development of MR-compatible therapy needles now allows these interventions to be performed under MR imaging guidance. Many studies have demonstrated the feasibility and effectiveness of this approach [14,15,16,17,18].

MRI-guided interventions as a modality for pain management have many advantages due to the following characteristics: (a) multiplanar imaging for better evaluation of the access route and control of needle position, and for complete visualization of the needle in non-orthogonal puncture directions due to the possibility of arbitrary orientation of the image planes in space; (b) selective T1 or T2 weighting with or without fat suppression for physiological tissue characterization, for example to visualize inflammatory processes for targeted injection; (c) the identification of vessels and injectates without the use of contrast agents in patients with hypersensitivity or by using non-iodine containing contrast agents in patients with impaired renal function for the intra-interventional visualization of important structures, such as the ureters; and (d) the avoidance of ionizing radiation for patients and physicians [18].

Computed tomography offers high temporal and spatial resolution, enabling fast and accurate needle positioning within the neuroforamen. This facilitates the effective perineural delivery of injectants, as supported by numerous prior studies [19,20,21]. However, a key limitation of both CT and fluoroscopy is the unavoidable exposure to ionizing radiation, affecting both patients and operators. This concern is especially critical in younger individuals or in cases requiring repeated interventions. Additionally, the routine reliance on contrast agents in these techniques presents further drawbacks, particularly for patients with allergies or impaired renal function [13,21,22].

Several available interventional studies use MRI at different magnetic field strengths for the needle navigation of PRT.

Scheffler et al. [23] demonstrated that lumbar nerve root infiltrations can be safely and effectively performed under 3 T MRI guidance without radiation exposure or iodinated contrast, achieving high targeting accuracy and excellent image quality. Their optimized workflow enabled procedural times comparable to conventional techniques, even in patients with challenging anatomy.

Guillemin et al. [24] showed that MRI-guided nerve root infiltration is a feasible, safe, and effective treatment option for chronic back pain, with precision comparable to CT-guided procedures. They highlighted the potential of improved needle visualization and suggested further studies on metal artifact reduction sequences to enhance procedural accuracy.

Sequeiros et al. [25] performed 60 of 61 successful injections using a 0.23 T open-MRI scanner, their results show that MRI guidance is accurate and safe in performing nerve root infiltration at lumbosacral area. The results of radicular pain relief from nerve root infiltration are comparable to CT or fluoroscopy studies on the subject.

Fritz et al. [26] study showed that freehand real-time MRI-guided lumbar spinal injection procedures are feasible, accurate, and safe when performed with a clinical—1.5 T MRI system.

Streitparth et al. [14] found that magnetic resonance fluoroscopy guided periradicular injection therapy for the lumbosacral spine, conducted under open 1.0 T MRI guidance, is accurate, safe, and efficient for relieving radicular pain symptoms.

Our study is the only study using phantom and 0.55 T MR-field with Magnetom Free.Max with an 80 cm scanner opening. Low-field MRI systems present several distinct benefits over their high-field counterparts. Because magnetic susceptibility effects scale with field strength, lower field strengths result in fewer susceptibility artifacts—an important advantage when visualizing structures near metallic instruments like needles. While much of MRI development has historically focused on increasing field strength, recent advances are driving renewed interest in low-field imaging. This resurgence is supported by improvements in image quality enabled by artificial intelligence, better field uniformity, and enhancements in system hardware. Notably, the integration of more efficient magnets and coil designs, along with helium-free operation, contributes to lower operational costs. Additionally, emerging deep learning-based image reconstruction techniques are making low-field MRI systems increasingly capable and economically viable [27,28,29].

Moreover, the availability of wider bore size reaching up to 80 cm enhances the feasibility and flexibility of conducting MRI-guided interventions by improving patient access and operator maneuverability within the scanner. This feature is particularly significant, as it allows for accommodating patients comfortably and facilitates the execution of a broader range of interventional techniques with improved accessibility and maneuverability within the scanner bore [30].

In our study, we achieved comparable accuracy comparable with those of the aforementioned studies. An accurate needle placement was achieved in 100% of the performed injections. These results are consistent with those reported in studies utilizing fluoroscopy or CT-guided techniques.

Additionally, our study highlights the significance of time and cost considerations using MRI within the context of low-field MRI systems. The utilization of low-field MRI systems, such as the Magnetom Free.Max, demonstrates substantial advantages in addressing these concerns. In instances where existing infrastructure poses challenges to MRI implementation, the Magnetom Free.Max seamlessly integrates into helium-free infrastructures. Furthermore, this system enhances accessibility by making MRI more affordable where it was previously unattainable. The absence of helium loss reduces costs, while the larger bore size improves accessibility and maneuverability, making the Magnetom Free.Max system a promising solution for precise and economical MRI-guided interventions. These findings underscore the potential of low-field MRI to advance interventional radiology practices and contribute to ongoing developments in medical imaging technology. Moreover, the optimization of the scan protocol could further reduce the intervention time, enhancing the overall efficiency of MRI-guided procedures.

The translation of MR-guided periradicular therapy using a low-field 0.55 T system into clinical practice has several workflow implications. The larger bore size of the system facilitates patient positioning and operator access, which is beneficial for interventional procedures. However, the total procedure time—particularly the duration of planning and imaging sequences—was significantly longer than with CT-based methods. This extended duration may limit throughput and requires careful scheduling in busy imaging departments. Additionally, MR-guided interventions demand close coordination between radiologists and MR technologists, as well as the availability of MR-compatible equipment and sterile technique adaptations suitable for the MRI environment. Optimizing sequence protocols and streamlining planning steps could help reduce intervention time and improve efficiency in future clinical applications.

While the current study focused on evaluating the technical feasibility and accuracy of different guidance systems, it did not include a formal analysis of the learning curve. This aspect is particularly relevant for MR-guided procedures, where operator familiarity with the imaging environment and workflow may influence performance. Future studies in clinical settings should aim to assess training effects over time and quantify the learning curve, especially when introducing MR-guided interventions into routine practice.

A notable limitation of this study is the use of a static phantom composed of homogeneous material, which does not fully reflect the complex characteristics of human tissues. Unlike in real patients, factors such as tissue compressibility, vascularity, and physiological motion—particularly respiratory movement or subtle shifts due to discomfort—are not represented. These aspects may influence needle handling and imaging in clinical practice. Nevertheless, the phantom provides a valuable, standardized platform for assessing the technical feasibility of the procedure under reproducible conditions.

In addition to the limitations of the phantom model regarding tissue characteristics, it is important to note that this study did not assess several clinically relevant aspects, including the accuracy of medication delivery, needle stability during respiratory motion, or the occurrence of procedural complications. These factors play a significant role in the clinical effectiveness and safety of periradicular therapy but cannot be adequately simulated in a static phantom. Future studies in live patient settings are warranted to explore these aspects and validate the clinical applicability of the presented MR-guided approach.

In conclusion, real-time MRI-guided lumbosacral periradicular injection therapy utilizing a 0.55 T MRI system is feasible with similar puncture times to CT guidance but needs more total time including planning images.

## Figures and Tables

**Figure 1 diagnostics-15-01413-f001:**
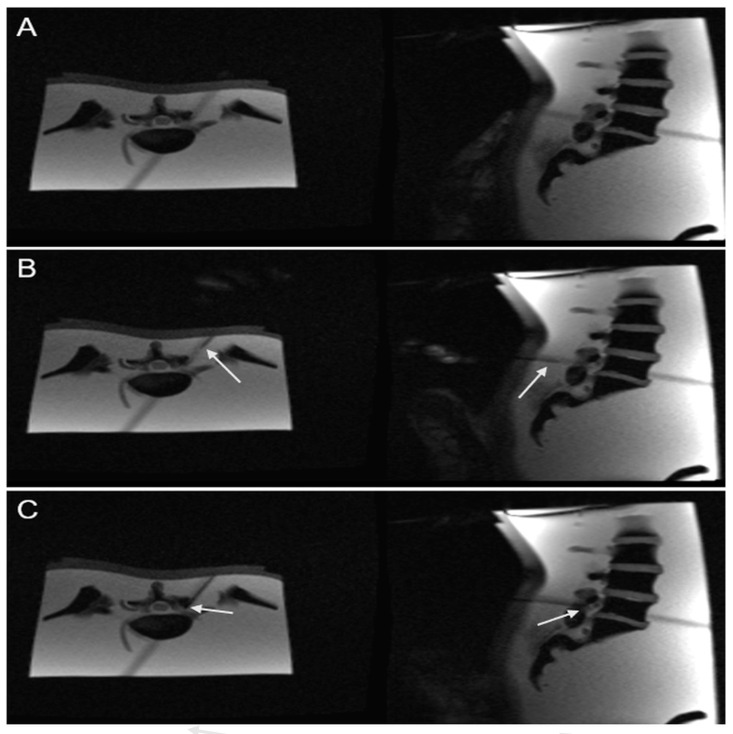
BEAT interactive sequence in (**A**–**C**). A shows the operator determining the exact skin entry point using the finger-pointing technique. The (**B**) shows the needle during the puncture, the needle on the way to the target. While (**C**) shows the needle at the target at the periradicular nerve root. In (**B**,**C**), the needle—particularly the tip—is marked with arrows to highlight its position in both axial and sagittal planes.

**Figure 2 diagnostics-15-01413-f002:**
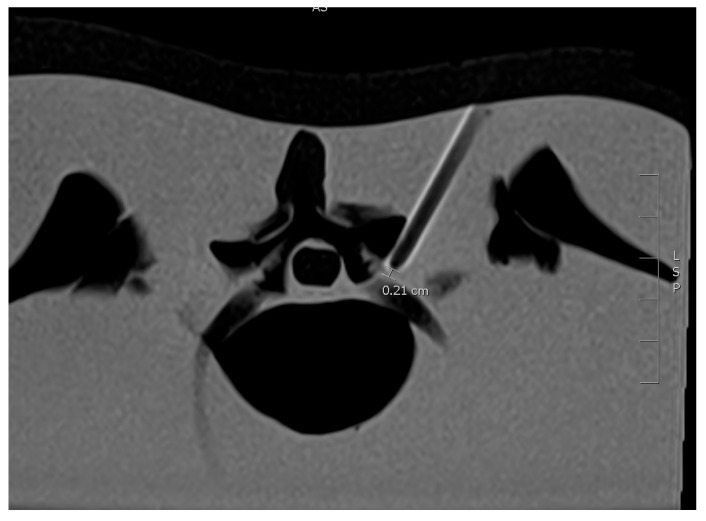
T2-weighted turbo spin echo (TSE) axial confirmation of the needle tip position ca. 2 mm to the nerve root.

**Figure 3 diagnostics-15-01413-f003:**
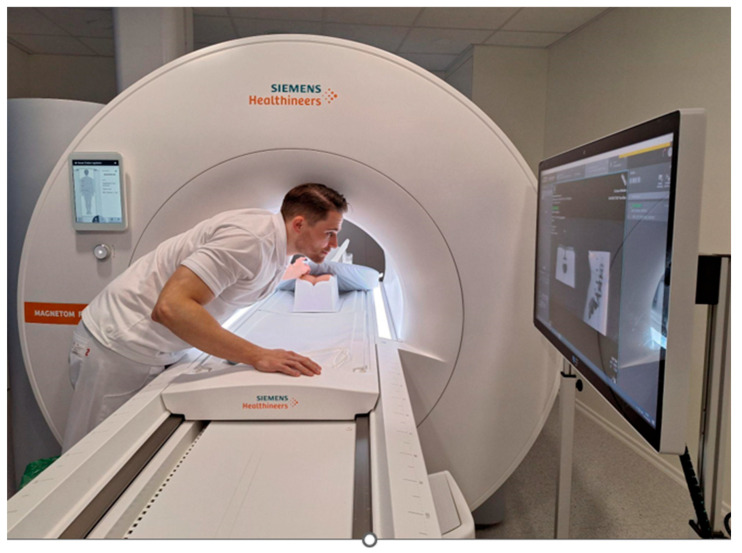
The operator performing a PRT in a 0.55 Tesla whole-body MRI system using a monitor in the room to access real-time imaging.

**Figure 4 diagnostics-15-01413-f004:**
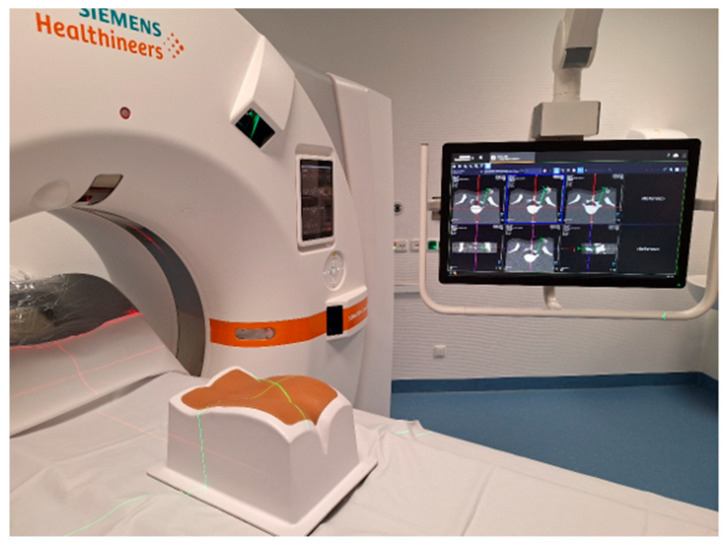
CT gantry equipped with four laser projectors attached to its housing is used to display the intended needle trajectory. This visualization is achieved through the projection of intersecting fan beam lasers onto a phantom.

**Figure 5 diagnostics-15-01413-f005:**
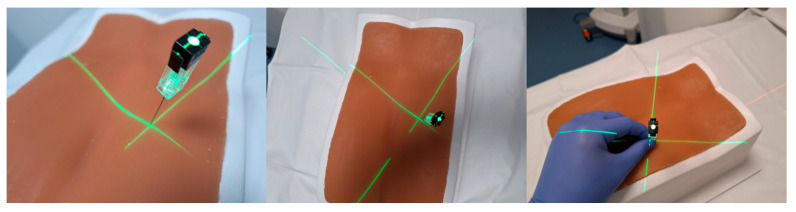
The phantom is illuminated with fan beam laser projection.

**Figure 6 diagnostics-15-01413-f006:**
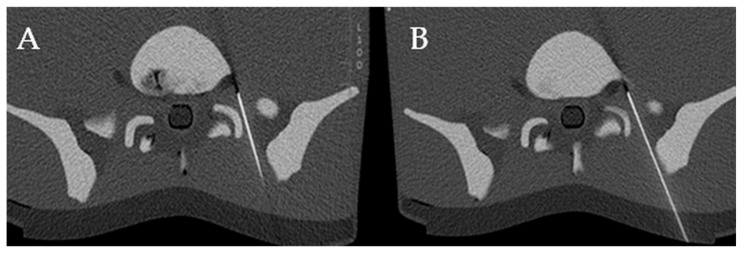
Axial CT images with conventional needle guidance (**A**) and using the 3D laser MDCT guidance system (**B**), confirming the needle tip position at the nerve root.

**Figure 7 diagnostics-15-01413-f007:**
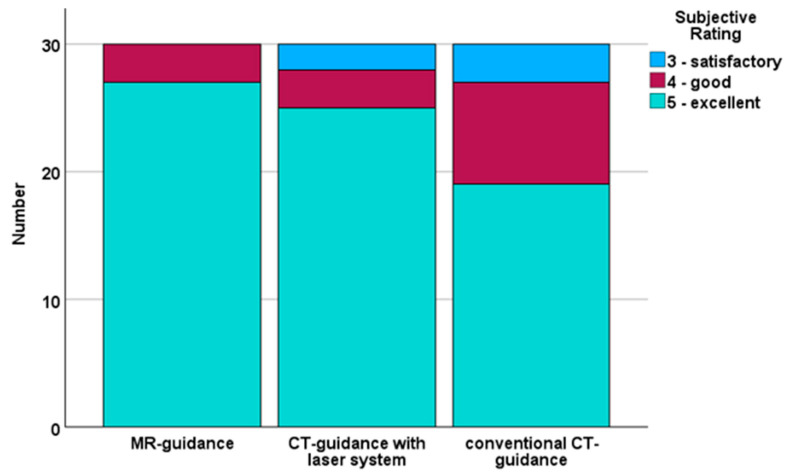
The subjective evaluations for all three guidance systems, with a median score of 5.

**Figure 8 diagnostics-15-01413-f008:**
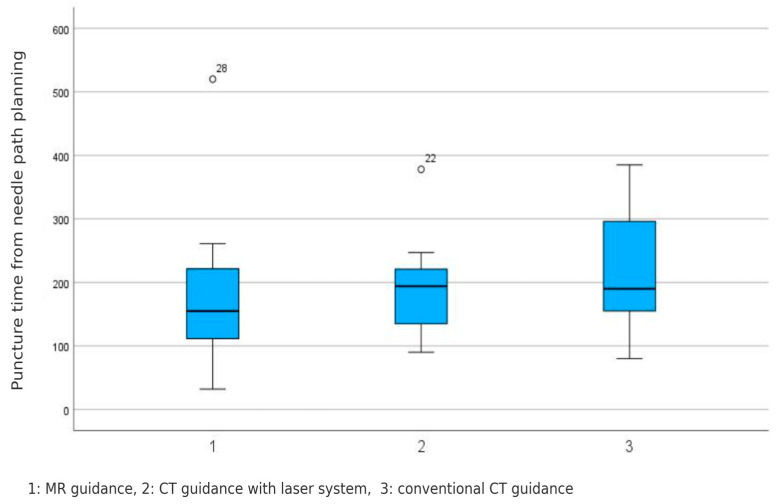
No significant difference in pure puncture time among the three guidance systems was registered. MR-guided: 178 ± 117 s, CT-guided with laser system: 186 ± 73 s, and conventional CT-guided: 218 ± 91 s (*p* = 0.482).

**Figure 9 diagnostics-15-01413-f009:**
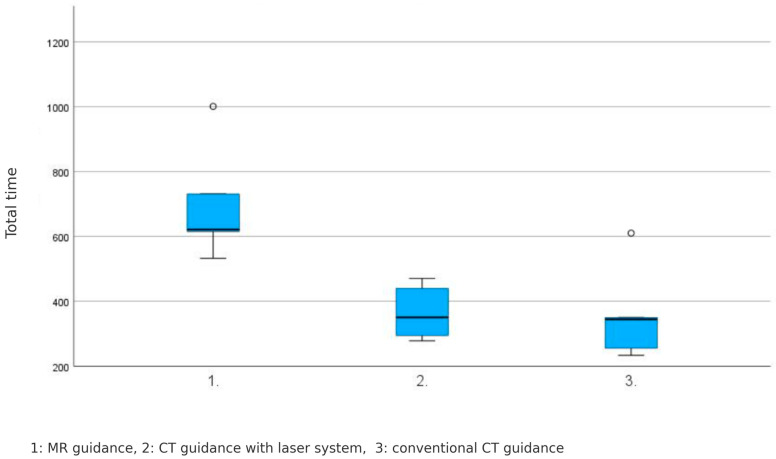
The total procedure time among the three guidance systems, with MR-guided PRTs taking significantly longer than the other two systems. MR-guided: 700 ± 182 s, CT-guided with laser system: 366 ± 85 s, und conventional CT-guided: 358 ± 150 s (*p* = 0.012).

## Data Availability

The raw data supporting the conclusions of this article will be made available by the authors on request.

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
