# Peer review of "Real-Time MR-Guided Lumbosacral Periradicular Injection Therapy Using a 0.55 T MRI System: A Phantom Study"

_diagnostics, 2025, doi:10.3390/diagnostics15111413_

Round 1

Reviewer 1 Report

Comments and Suggestions for Authors

The study addresses an important and emerging topic: the feasibility of low-field (0.55 T) MRI-guided periradicular therapy using real-time imaging. The design (phantom model, multi-operator performance) is methodologically acceptable. Here are my comments;

  1. Human tissues differ in compressibility, vascularity, and movement (respiratory motion, patient discomfort).
  2. the study does not analyze the learning curve
  3. A power analysis should be presented to justify sample size.
  4. No assessment of accuracy in medication delivery, needle stability during respiration, or overall procedural complications
  5. Discuss real-world workflow implications
  6. Some sections in the Discussion repeat points. Please, avoid redundancy.

Author Response

Reviewer Comment 1:
Human tissues differ in compressibility, vascularity, and movement (respiratory motion, patient discomfort).

Author Response 1:
We thank the reviewer for this important observation. In response, we have revised the Limitations section to acknowledge these differences more clearly. The revised paragraph now reads: “A notable limitation of this study is the use of a static phantom composed of homogeneous material, which does not fully reflect the complex characteristics of human tissues. Unlike in real patients, factors such as tissue compressibility, vascularity, and physiological motion—particularly respiratory movement or subtle shifts due to discomfort—are not represented. These aspects may influence needle handling and imaging in clinical practice. Nevertheless, the phantom provides a valuable, standardized platform for assessing the technical feasibility of the procedure under reproducible conditions.”

We hope this addition adequately addresses the reviewer’s concern.

Reviewer Comment2:
The study does not analyze the learning curve.

Author Response2:
We thank the reviewer for this valuable comment. We agree that analyzing the learning curve would provide important insights, particularly given the novelty of MR-guided PRT using a low-field system. However, this was beyond the scope of the current study, which was designed to evaluate technical feasibility and procedural accuracy in a controlled setting. Nonetheless, we acknowledge this as a relevant aspect for future clinical studies, and we have now added a statement to the Discussion section accordingly.

Added to Discussion:
While the current study focused on evaluating the technical feasibility and accuracy of different guidance systems, it did not include a formal analysis of the learning curve. This aspect is particularly relevant for MR-guided procedures, where operator familiarity with the imaging environment and workflow may influence performance. Future studies in clinical settings should aim to assess training effects over time and quantify the learning curve, especially when introducing MR-guided interventions into routine practice.

Reviewer Comment 3:
A power analysis should be presented to justify sample size.

Author Response 3:
We thank the reviewer for this important suggestion. The primary aim of this study was to assess the technical feasibility of MR-guided periradicular therapy using a low-field system in a phantom model. As such, the study design was exploratory in nature and not powered to detect small differences between guidance methods. Given the pilot character and the use of a fixed number of procedures across guidance modalities, a formal power calculation was not performed. We acknowledge this as a limitation and have clarified it in the manuscript accordingly.

Added to the study design:
As this was a feasibility study with a predefined number of procedures and operators, no formal power analysis was conducted. The study was not designed to detect statistically significant differences with high power but rather to provide an initial comparative evaluation of guidance modalities in a controlled phantom setting. Future studies with larger sample sizes and formal sample size estimation will be needed to confirm these findings and explore subtle performance differences.

Reviewer Comment 4:

No assessment of accuracy in medication delivery, needle stability during respiration, or overall procedural complications.

Author Response 4:

We appreciate the reviewer’s insightful comment. As this study was conducted on a static phantom model, certain clinically relevant parameters—such as medication distribution accuracy, needle displacement during respiration, and procedure-related complications—could not be assessed. We have now clarified this limitation in the manuscript and recognize that these aspects are crucial in translating technical feasibility into clinical practice. Future in vivo studies are needed to evaluate these parameters under real physiological conditions.

Added to Limitations:

In addition to the limitations of the phantom model regarding tissue characteristics, it is important to note that this study did not assess several clinically relevant aspects, including the accuracy of medication delivery, needle stability during respiratory motion, or the occurrence of procedural complications. These factors play a significant role in the clinical effectiveness and safety of periradicular therapy but cannot be adequately simulated in a static phantom. Future studies in live patient settings are warranted to explore these aspects and validate the clinical applicability of the presented MR-guided approach.

Reviewer Comment 5:

Discuss real-world workflow implications.

Author Response 5:

We thank the reviewer for this important comment. We agree that translating technical feasibility into clinical routine requires consideration of practical workflow factors. We have now added a paragraph to the Discussion addressing real-world implications such as room setup, team coordination, and scan time management when integrating low-field MRI-guided interventions into clinical practice.

Added to Discussion :

The translation of MR-guided periradicular therapy using a low-field 0.55 T system into clinical practice has several workflow implications. The larger bore size of the system facilitates patient positioning and operator access, which is beneficial for interventional procedures. However, the total procedure time—particularly the duration of planning and imaging sequences—was significantly longer than with CT-based methods. This extended duration may limit throughput and requires careful scheduling in busy imaging departments. Additionally, MR-guided interventions demand close coordination between radiologists and MR technologists, as well as the availability of MR-compatible equipment and sterile technique adaptations suitable for the MRI environment. Optimizing sequence protocols and streamlining planning steps could help reduce intervention time and improve efficiency in future clinical applications.

Reviewer Comment 6:

Some sections in the Discussion repeat points. Please, avoid redundancy.

Author Response 6:

We appreciate the reviewer’s observation regarding redundancy in the Discussion section. We have carefully reviewed and revised the text to remove repetitive statements, particularly those relating to technical feasibility, advantages of low-field MRI, and the comparison with CT-guided procedures. We hope this improves the clarity and readability of the manuscript.

Reviewer 2 Report

Comments and Suggestions for Authors

The article evaluate the feasibility of MRI-guided periradicular nerve root injection therapy  using a 0.55-T magnetic resonance imaging (MRI) system in a phantom. It is a well writing manuscript. There is still something need to be verified. 

1.In the Figure 1, please show the needle by using arrow

2. Please using the MRI picture to show the distance from the needle tip to the nerve root. also adding the measurement unit.

3. Please adding the picture like Figure 2 to show the 3D laser guidance system and conventional CT guidance procedure.

4. Please  explain what is the quality of of needle position, and what is the satisfactary good and excellent.

Author Response

Reviewer Comment 1:
In Figure 1, please show the needle by using an arrow.

Author Response1:
We thank the reviewer for the helpful suggestion. We have now added white arrows in Figure 1 to indicate the needle, particularly the tip, in both axial and sagittal planes in panels B and C. This has also been explicitly described in the revised figure legend as follows:
“In B and C, the needle—particularly the tip—is marked with red arrows to highlight its position in both axial and sagittal planes.”

Reviewer Comment 2:
Please use the MRI image to show the distance from the needle tip to the nerve root, also adding the measurement unit.

Author Response 2:
We thank the reviewer for this constructive comment. In response, we have updated Figure 2 to include a visual indication of the distance between the needle tip and the adjacent nerve root, along with an approximate measurement in millimeters. This is now reflected in the revised figure legend, which reads:
“Figure 2. T2-weighted turbo spin echo (TSE) axial confirmation of the needle tip position, approximately 2 mm from the periradicular nerve root.”

Reviewer Comment 3:
Please add an image similar to Figure 2 to show the 3D laser guidance system and conventional CT guidance procedure.

Author Response 3:
We thank the reviewer for this helpful suggestion. In response, we have added Figure 6, which displays representative axial images from the conventional CT-guided and 3D laser guidance system procedures. This figure illustrates needle positioning relative to anatomical landmarks and the nerve root.

Reviewer Comment 4:
Please explain what is the quality of needle position, and what is satisfactory, good, and excellent.

Author Response 4:
We thank the reviewer for this important question. The quality of needle position was evaluated by two experienced neuroradiologists using a 5-point ordinal scale based on proximity of the needle tip to the target nerve root and its orientation within the expected anatomical pathway. The grading was defined as follows:

  • Excellent (5): Precise needle placement in direct contact with the target nerve root in optimal trajectory.

  • Very Good to Good (4): Close needle positioning (<2 mm from the nerve root) with correct orientation and minor deviation from ideal path.

  • Satisfactory (3): Acceptable needle placement (within 3–5 mm of target) with slight angulation but still allowing effective therapy.

  • Bad (2): Suboptimal needle position requiring significant correction or unlikely to deliver medication effectively.

  • Very Bad(1): Misplacement or failed targeting.

This subjective scoring reflected both anatomical accuracy and technical adequacy for periradicular injection therapy.

Reviewer 3 Report

Comments and Suggestions for Authors

The manuscript titled “Real-Time MR-Guided Lumbosacral Periradicular Injection Therapy Using a 0.55 T MRI System: A Phantom Study” evaluates the feasibility and accuracy of using a low-field 0.55 T MRI system for guiding lumbosacral periradicular injections. Five radiologists with varying experience levels performed procedures on a lumbar spine phantom under three guidance conditions: MR-guided, CT-guided with laser navigation, and conventional CT-guided. Results showed comparable puncture accuracy across all modalities, but MR-guided procedures required significantly longer total intervention time due to imaging and planning sequences. The study supports the potential of low-field MRI systems for radiation-free, real-time interventional spine procedures, although phantom-based findings limit direct clinical extrapolation.

Comments & Questions:

  1. Time Efficiency and Workflow Optimization
    The MR-guided intervention took about twice the time compared to CT-guided methods. Could the authors suggest specific protocol optimizations (e.g., faster planning sequences, abbreviated imaging) that could realistically shorten MR procedure time in a clinical setting?
  2. Needle Visualization in Low-Field MRI
    The study discusses reduced susceptibility artifacts in low-field MRI. However, were there any difficulties encountered in needle tip visualization or trajectory monitoring during real-time imaging?
  3. Phantom Limitations and Motion Simulation
    Since the phantom lacks tissue heterogeneity and motion simulation, would the authors consider using a dynamic phantom model or adding controlled motion in future experiments to better mimic clinical conditions?

Author Response

Reviewer Comment1:
Time Efficiency and Workflow Optimization: The MR-guided intervention took about twice the time compared to CT-guided methods. Could the authors suggest specific protocol optimizations (e.g., faster planning sequences, abbreviated imaging) that could realistically shorten MR procedure time in a clinical setting?

Author Response1:
We thank the reviewer for this important point. The longer procedure time in MR-guided interventions is indeed a limitation. However, in clinical practice, workflow can be optimized using faster planning sequences (e.g., compressed sensing), reducing scan coverage, and minimizing averaging in dynamic imaging. In our ongoing clinical use of MR-guided interventions, we frequently utilize a relatively fast VIBE sequence (~16 seconds) to verify needle position. Further protocol refinements and semi-automated planning tools may help reduce MR-guided procedure time to levels comparable with CT-guided techniques.

Comment 2:
Needle Visualization in Low-Field MRI: The study discusses reduced susceptibility artifacts in low-field MRI. However, were there any difficulties encountered in needle tip visualization or trajectory monitoring during real-time imaging?

Author Response 2:

We thank the reviewer for this thoughtful question. In our experience, needle visualization in the low-field (0.55 T) MRI system was very good. The reduced susceptibility artifacts allowed clear depiction of both the needle shaft and tip. Real-time sequences such as BEAT IRT provided reliable trajectory monitoring during insertion to confirm the needle position. Overall, we did not encounter significant difficulties in needle visualization or trajectory control during the procedures.

Comment 3:
Phantom Limitations and Motion Simulation: Since the phantom lacks tissue heterogeneity and motion simulation, would the authors consider using a dynamic phantom model or adding controlled motion in future experiments to better mimic clinical conditions?

Author Response 3:
We thank the reviewer for this valuable suggestion. While incorporating motion simulation in future experimental setups would indeed enhance realism, we have already moved forward with clinical implementation. To date, we have performed approximately 18 MR-guided interventions on patients using the 0.55 T system and are very satisfied with the image quality, workflow, and clinical feasibility. These early patient experiences further support the promising results observed in our phantom study.

Round 2

Reviewer 1 Report

Comments and Suggestions for Authors

In its current form, the manuscript is suitable for publication.

Author Response

Reviewer Comment:
In its current form, the manuscript is suitable for publication.

Author Response:
We sincerely thank the reviewer for the positive evaluation and supportive feedback.